# The antigenic switching network of *Plasmodium falciparum* and its implications for the immuno-epidemiology of malaria

Robert Noble[1†], Zóe Christodoulou[2†], Sue Kyes[2], Robert Pinches[2], Chris I Newbold[2*], Mario Recker[1*]

[1]Department of Zoology, University of Oxford, Oxford, United Kingdom; [2]Weatherall Institute of Molecular Medicine, John Radcliffe Hospital, Oxford, United Kingdom

**Abstract** Antigenic variation in the human malaria parasite *Plasmodium falciparum* involves sequential and mutually exclusive expression of members of the *var* multi-gene family and appears to follow a non-random pattern. In this study, using a detailed in vitro gene transcription analysis of the culture-adapted HB3 strain of *P. falciparum*, we show that antigenic switching is governed by a global activation hierarchy favouring short and highly diverse genes in central chromosomal location. Longer and more conserved genes, which have previously been associated with severe infection in immunologically naive hosts, are rarely activated, however, implying an in vivo fitness advantage possibly through adhesion-dependent survival rates. We further show that a gene's activation rate is positively associated sequence diversity, which could offer important new insights into the evolution and maintenance of antigenic diversity in *P. falciparum* malaria.

*For correspondence: chris.
newbold@imm.ox.ac.uk (CIN);
mario.recker@zoo.ox.ac.uk (MR)

†These authors contributed
equally to this work

Competing interests: The
authors declare that no
competing interests exist.

Reviewing editor: Mercedes
Pascual, University of Michigan,
United States

## Introduction

Acquired antibody-mediated immunity against *Plasmodium falciparum* is directed against the disease-causing, intra-erythrocytic life-stage of the parasite's life-cycle. Continual exposure to *P. falciparum* infection can lead to a form of semi-immunity, whereby protection against life-threatening disease is acquired after a few infections only (*Gupta et al., 1999*) whereas sterile and long-lasting immunity against infection is never achieved through natural exposure (*Greenwood et al., 1987*; *Marsh, 1992*; *Tran et al., 2013*). The process of natural acquired immunity therefore coincides with a transition from acute and severe infection in young children to asymptomatic carriage in older individuals (*Marsh and Snow, 1997*). At the core of this variable outcome of infection and poor development of immunity lies the major parasite virulence factor and variant antigen, *P. falciparum* erythrocyte membrane protein 1 (PfEMP1). Members of this family of proteins are expressed on the surface of infected red blood cells and are involved in the binding of parasitised cells to other host cells and tissues (for an overview see *Kraemer and Smith, (2006)*). The conferred binding phenotypes are believed to enhance parasite survival by avoiding splenic clearance (*Boone and Watters, 1995*; *Buffet et al., 2009*) and also contribute to malaria pathology through parasite sequestration in the deep vasculature (*Ockenhouse et al., 1991*; *Pongponratn et al., 1991*). Mutually exclusive transcriptional switching between PfEMP1 variants during the course of an infection thus affords the parasite an immune evasion strategy and a means to exploit different host tissues (*Roberts et al., 1992*; *Smith et al., 1995*).

Each haploid parasite genome contains approximately 60 members of the *var* gene family (*Gardner et al., 2002*) encoding distinct PfEMP1 variants, with a high degree of divergence between the antigenic repertoires of any two parasites. The high diversity of *var* genes and *var* gene repertoires is predominantly

**eLife digest** Our ability to acquire immunity to a disease depends on our immune system learning to recognise foreign molecules—called antigens—that are specific to the disease-causing virus, bacterium or parasite. However, some pathogens, such as the malaria-causing parasite *Plasmodium falciparum*, get around this defence through a process called antigenic variation. This involves the parasite switching between different antigens over the course of an infection, preventing the host immune system from learning to recognise them and leading to infections that last many weeks or even months.

The main antigen in *P. falciparum* is a protein called PfEMP1, which is encoded by a family of genes called *var* ('variable'). *Var* genes have evolved to be highly diverse, and different parasites have different repertoires of around 50–60 *var* genes. This ensures that there are a huge number of distinct variants of the PfEMP1 antigen available within the population, allowing the malaria parasite to maintain long-lasting infections and also to infect the same individuals again and again.

Previous work has shown that the expression of *var* genes is not random, but it is not clear what determines which genes are expressed at any given time. Now, Noble et al. have performed a detailed investigation of antigenic switching in *P. falciparum*. Using clonal parasites, they closely monitored the expression of the entire *var* gene repertoire during many generations of parasite culture. They observed that although different cultures initially expressed distinct *var* genes, most of them ended up expressing two particular genes—*var27* and *var29*—at high levels, indicating a hard-wired gene 'activation hierarchy'.

Noble et al. found that whenever the parasites switched antigens, *var* genes that were centrally located on chromosomes—such as *var27* and *var29*—were more likely to be activated than those at the ends of chromosomes. Moreover, *var* genes that were highly diverse were more likely to be activated than more conserved genes: this is the first evidence linking *var* gene evolution with gene activation probabilities. Together, these factors gave rise to the proposed activation hierarchy, which favours genes optimised for immune evasion and aids their continued evolution and diversification. Further work is now needed to identify the molecular mechanisms that control antigenic switching and to determine whether these could represent new therapeutic targets for malaria.

generated and maintained by frequent recombination and gene conversion events (*Deitsch et al., 1999*; *Ward et al., 1999*; *Freitas-Junior et al., 2000*; *Taylor et al., 2000*; *Kraemer et al., 2007*; *Frank et al., 2008*), which has also resulted in a high degree of mosaicism within these genes (*Ward et al., 1999*; *Taylor et al., 2000*; *Bull et al., 2008*). The modular and highly polymorphic, extracellular portion of PfEMP1 is composed of a variable number of Duffy-binding-like domains (DBL; at least two and up to seven per gene) and cysteine-rich interdomain regions (CIDR; up to two per gene) (*Kraemer et al., 2007*). Despite the overall diversity of *var* genes at the sequence level, distinct associations with severe malaria have been identified for several such genes (*Salanti et al., 2003*; *Jensen et al., 2004*), with the involvement of *var2csa* in pregnancy-associated malaria being the clearest example to date (*Salanti et al., 2004*), as well as associations with particular DBL domains (*Warimwe et al., 2009*) and combinations of DBL and CIDR (*Avril et al., 2012*; *Claessens et al., 2012*; *Lavstsen et al., 2012*).

*Var* genes can also be classified according to a conserved upstream promoter sequence into four types termed UpsA, UpsB, UpsC, and UpsE (*Lavstsen et al., 2003*). Of particular interest are the UpsA-type genes, which appear to have diverged from the other groups (*Kraemer et al., 2007*; *Bull et al., 2008*) and have been shown to be upregulated during severe infections in young hosts (*Jensen et al., 2004*; *Bull et al., 2005*; *Kaestli et al., 2006*; *Kyriacou et al., 2006*; *Rottmann et al., 2006*; *Warimwe et al., 2009*). As a possible consequence, acquisition of anti-PfEMP1 immunity appears to follow a particular order, with UpsA variants generally being the first to be broadly recognised in older children (*Warimwe et al., 2009*; *Cham et al., 2010*). Nevertheless, clear associations between the expression of particular genes or Ups-groups, host age and severity of disease are still missing and could in fact be strain- or location-specific. Understanding the underlying pattern of antigenic switching between *var* genes is therefore important to explain not only the mechanisms and dynamics of persistent

infections but also the relationship between immune-mediated expression of particular PfEMP1 sub-types and infection outcomes.

Various studies have investigated *var* gene expression during natural or experimental infections (*Peters et al., 2002*; *Lavstsen et al., 2005*; *Wunderlich et al., 2005*; *Blomqvist et al., 2010*). However, as the diversity and order of PfEMP1 variants in the human host is influenced by immune responses and other host factors (*Kyriacou et al., 2006*; *Warimwe et al., 2009*), attempts to elucidate the underlying patterns of antigenic change and estimate related switching parameters are commonly based on in vitro cultured parasites in the absence of selection pressure. In this setting, Horrocks et al. (*Horrocks et al., 2004*) found that the rates at which *var* genes activate or deactivate are non-random, gene-specific and highly dissimilar. It has furthermore been suggested that *var* genes occupying subtelomeric loci tend to switch off faster than those positioned in central chromosomal regions (*Frank et al., 2007*), offering a possible explanation for the rapid decline in the transcription of subtelomeric genes in parasites during culture adaptation (*Peters et al., 2007*; *Zhang et al., 2011*). More recently, *Recker et al. (2011)* proposed that *var* gene transcriptional change involves a highly structured pathway that has evolved in response to a trade-off between the within-host and the between-host level fitness. However, most of these previous studies were limited in analytical depth due to restrictions in the number of variants considered and/or time points at which gene transcription levels were measured, and could therefore describe only small fragments of the whole switching network.

In this study, we used a novel approach to provide the first detailed characterisation of antigenic switching in the HB3 isolate of *P. falciparum*. We analysed pooled *var* gene transcription data from verified quantitative real time PCR measurements of a diverse set of clonal parasite cultures using a statistically rigorous method of parameter estimation. This revealed a global hierarchy in *var* gene activation favouring a highly diverse set of genes in central chromosomal locations. Our results further suggest a role of active gene transcription in the generation of antigenic diversity and will have important implications for understanding the age- and exposure-dependent pathology of *P. falciparum* malaria.

## Results

11 subclones of the lab-adapted HB3 isolate of *P. falciparum* were obtained by serial dilution of the parent culture. The clones were cultivated and transcript levels of 38 *var* genes measured by quantitative PCR, first at 29–33 parasite generations (8–9 weeks) after cloning and then at several time points over the following 60–79 generations in each culture (*Table 1*); a detailed description can be found in the 'Materials and methods' section. These 38 genes comprised almost all known functional HB3 *var* genes (*Kraemer et al., 2007*; *Rowe et al., 2009*) and one partial gene. The HB3 parent culture was also observed at seven time points. At the first observation, each of the 11 clones predominantly transcribed one of the nine different *var* genes, which were most likely the genes transcribed at the time of cloning. Importantly, these nine genes—hereafter referred to as 'starter genes'—were representative of the *var* repertoire diversity in terms of chromosomal location and Ups type.

### *Var* gene transcription dynamics

The proportional transcript levels of the different starter genes declined at notably different rates over the course of the experiment (*Figure 1*, *Figure 1—figure supplements 1 and 2*). For example, the centrally located, UpsC starter gene *var29* in culture #8 accounted for more than 90% of the total transcript level at the first measurement and remained the dominant variant over the following 60 generations. In contrast, the transcript level of the *var35* starter gene of culture #11 (subtelomeric, UpsA promoter type) was already below 50% of the total transcript level at the point of the first observation (approximately 30 generations post cloning) and continued to decline during the time course of culture, to be eventually replaced by other genes as dominant transcripts. Similar patterns were seen in cultures #3 and #7, which both had centrally located, UpsC starter genes, whereas the remaining five starter genes showed intermediate rates of decline.

Of particular interest was the apparent convergence of most cultures towards high transcription of *var27* and/or *var29* (*Figure 1*, *Figure 1—figure supplements 1 and 2*), which are both situated in central chromosomal location but have different promoter types (UpsB and UpsC, respectively). In light of the very different initial conditions, in terms of *var* gene transcription levels at which these cultures were established, this indicates that antigenic switching in *P. falciparum* might be governed by an activation hierarchy.

**Table 1.** Parasite culture and RNA sampling information

| Culture | Starter gene | Observed generations |
|---|---|---|
| 1 | var11 | 29, 48, 59, 69, 89 |
| 2 | var13 | 29, 48, 69, 89, 99, 108 |
| 3 | var13 | 30, 49, 70, 90, 100, 109 |
| 4 | var5 | 30, 39, 49, 60, 70, 90 |
| 5 | var27 | 30, 49, 70, 90 |
| 6 | var27 | 29, 38, 48, 69, 89 |
| 7 | var28 | 33, 52, 63, 73, 93 |
| 8 | var29 | 30, 49, 70, 80 |
| 9 | var30 | 29, 48, 59, 69, 89 |
| 10 | var31 | 29, 48, 59, 69, 89, 99, 108 |
| 11 | var35 | 32, 51, 65, 72, 92 |
| Parent | – | 0, 9, 19, 40, 60, 70, 79 |

To fully understand the patterns of antigenic switching underlying the observed transcriptional changes in our parasite cultures, we estimated relevant switch parameters using a previously described Markov chain Monte Carlo (MCMC) method (**Noble and Recker, 2012**) ('Materials and methods'). We have previously proposed that *var* gene switching can be fully described dynamically by two sets of parameters ('Materials and methods' and [**Recker et al., 2011**]). First, each gene $i$ has an off-rate, $\omega_i$, which is the per generation probability of switching from active to silenced. Second, each gene has a switch bias, $\beta_{ij}$, which is the probability that when gene $i$ switches off, gene $j$ becomes activated. We applied our algorithm to the full 38-gene data set and separately to a reduced set of the 16 most transcribed genes, and both models generally gave good and similar fits between the data and predicted transcript levels (**Figure 1—figure supplements 1 and 2**), with deviations resembling stochastic measurement errors.

Our method provided estimated switch parameters for the entire *var* repertoire, which are illustrated in **Figure 2** for the 16 most highly transcribed genes. Due to higher signal to noise ratios, parameter estimates of the starter genes (indicated in red) were more accurate and resulted in much narrower credible intervals, as indicated by less fuzzy rings that represent respective switch probabilities. As clearly demonstrated in **Figure 2**, there was wide distribution in switch biases and off-rates, and we next analysed for specific associations between these parameters and other gene characteristics.

## Off-rate estimates suggest a role of chromosomal location, not promoter type

Estimated off-rates ranged from 0.3% to more than 5% per generation, with a mean of approximately 2.9% (**Table 2**). Notably, the two most transcribed genes, *var29* and *var27* (both centrally located on chromosome 4), had exceptionally low off-rates of approximately 0.3% and 0.8%, respectively, whereas the other starter genes switched off at rates between 1.4% and 3.5% per generation. In general we found off-rates of centrally located genes to be significantly lower than of genes in subtelomeric regions (Welch's $t$ = 3.3, p=0.002), as shown in **Figure 3**. This result remained significant ($p = 0.035$) when we accounted for uncertainty in our estimates ('Materials and methods').

Previous studies reported that UpsA *var* genes from patient isolates were down-regulated faster than other *var* genes during adaptation to culture (**Peters et al., 2007**; **Zhang et al., 2011**). The HB3 repertoire contains seven UpsA genes, all but one of which are subtelomeric. We compared the off-rates of the subtelomeric UpsA genes with those of other subtelomeric genes and found no significant difference (Welch's $t$ = 0.65, p=0.54; accounting for uncertainty p=0.55), as shown in **Figure 3**. Among centrally located genes we compared UpsB with UpsC and again found no significant difference (**Figure 3**; Welch's $t$ = 0.44, p=0.44; accounting for uncertainty p=0.57). This lends support to a previous study suggesting that the rate at which transcriptionally active *var* genes become silenced is less dependent on promoter type than on chromosomal location (**Frank et al., 2007**).

## *Var* gene switching is governed by an activation hierarchy

In each of our cultures, gene transcription levels spanned several orders of magnitude, and only a minority of genes were ever highly transcribed during the experiment. As mentioned earlier, despite the diversity of the starter genes, the relative transcript levels of the activated genes appeared to converge towards similar levels across all cultures (**Figure 1**, **Figure 1—figure supplements 1 and 2**). To test the degree of similarity in gene activation, we excluded the nine starter genes and ranked the rest of the repertoire according to average transcript levels at later time points (more than 65 generations) for each culture. Pair-wise comparison showed that the ranking was highly correlated between

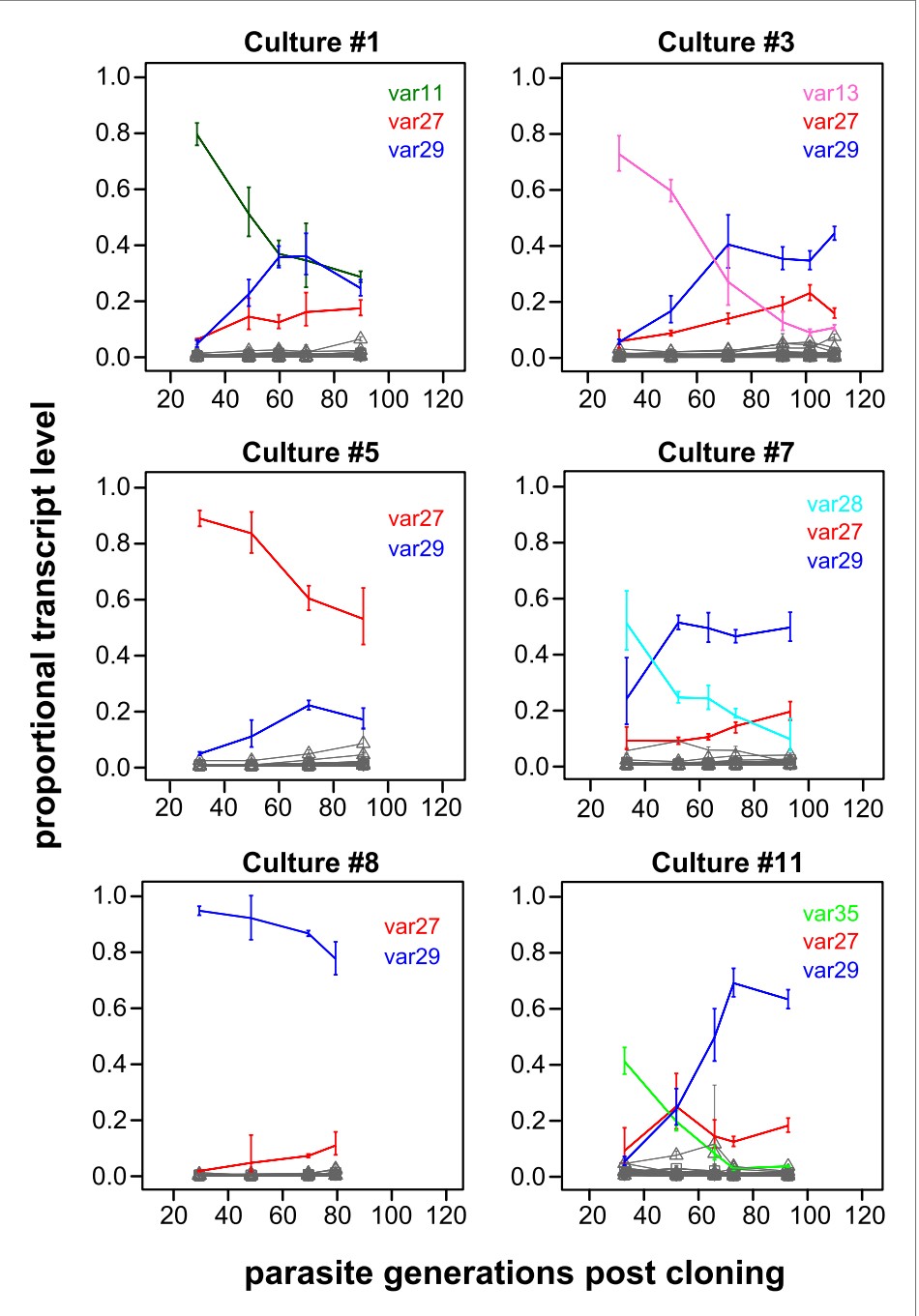

**Figure 1**. Proportional var transcript levels for six in vitro cultures. The parasite cultures initially expressed a variety of dominant 'starter genes', which switched off at notably different rates. Nevertheless, most cultures converged towards high level transcription of two centrally located genes *var27* and *var29* (red and blue lines, respectively), whereas most other gene transcripts (grey lines) remained at relatively low levels.

The following figure supplements are available for figure 1:

**Figure supplement 1**. Proportional *var* transcript levels and model output for cultures 1–6.

**Figure supplement 2**. Proportional *var* transcript levels and model output for cultures 7–11.

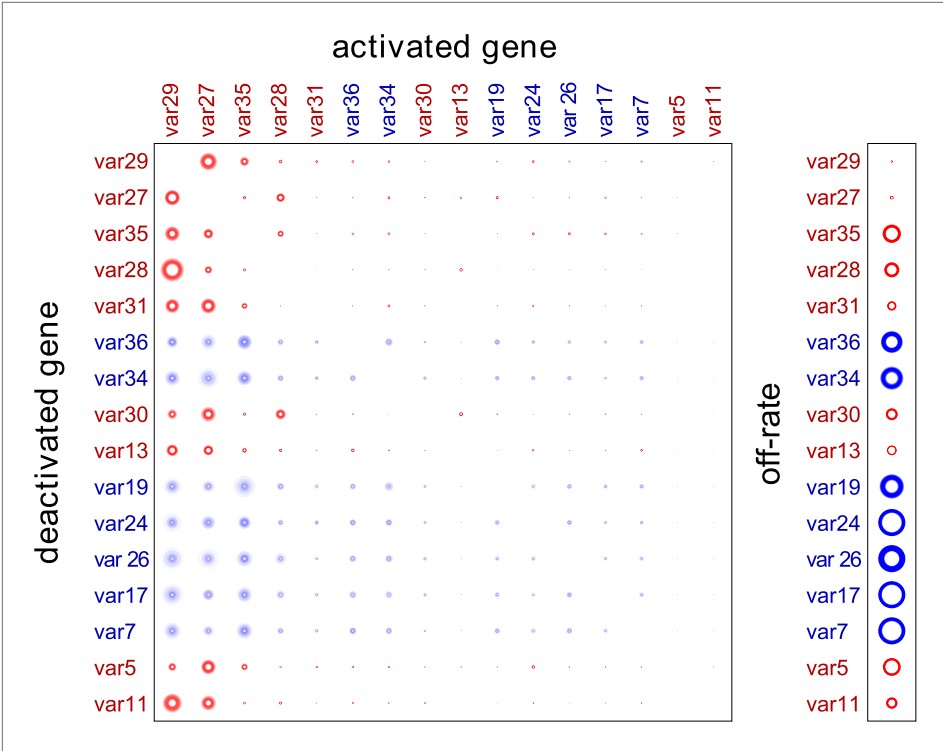

**Figure 2**. Estimation of switch parameters of highly transcribed genes. Parameter estimations for the 16 most transcribed *var* genes are represented as a switch matrix and an off-rate vector. The diameter of a circle in the *i*th row and *j*th column of the matrix is proportional to the switch bias $\beta_{ij}$ from gene *i* to gene *j*, and the diameter of a circle in the off-rate vector is proportional to the rate $\omega_i$ at which gene *i* becomes silenced. The fuzziness indicates uncertainty in the estimate, such that the darkness of each concentric ring is proportional to the likelihood that the parameter is within the corresponding range (**Noble and Recker, 2012**). Starter gene parameters, which are more precisely estimated, are in red, and genes are arranged from left to right, and top to bottom, in order of average transcript level.

The following source data and figure supplements are available for figure 2:

**Source data 1**. Estimated switch biases and 95% credible intervals (CI).

**Figure supplement 1**. Validation of estimated switching parameters.

the cultures (Spearman's $\rho > 0.8$, p=10$^{-5}$ for all pairs), suggesting that the outcome of antigenic switching in *P. falciparum*, or at least within the HB3 isolate, is largely governed by an activation hierarchy that is independent of the currently activated gene.

Consistent with this hypothesis, the estimated switch biases between genes, $\beta_{ij}$, were remarkably similar, so that $\beta_{ij} \approx \beta_{kj}$ for each *i*, *j* and *k*. (**Figure 2**, **Figure 2—source data 1**, **Figure 2—figure supplement 1**). All genes were found to have high switch biases towards the same few genes, which explain the observed convergence in *var* transcription in our cultures. As expected, the highest biases across the repertoire were found towards *var29* and *var27*, with some genes also frequently switching towards *var35* and/or *var28* (both UpsC-types in central chromosomal location).

To better describe the observed activation hierarchy, we used our MCMC method to estimate the average switch bias towards each gene, which we hereafter refer to as the gene's activation bias. That is, the activation bias of gene *j* is defined as the average of the switch biases $\beta_{ij}$ for all deactivating genes *i*. It can be understood as a gene specific, *per switch* activation probability and is therefore distinct from the previously defined *on-rate* (**Horrocks et al., 2004**), in that it is invariant to transcript levels. As shown in **Table 2**, there was a clear hierarchy in gene activation biases spanning over three orders of magnitude. The ranking also appeared to be non-random so that, for example, most genes within central chromosomal location were situated in the top half of the table. Analysing for possible associations with genetic attributes

**Table 2.** Parameter estimates for the HB3 *var* repertoire with 95% credible intervals

| Gene | Location | Domains* | Ups | Off-rate (95% CI) | Activation bias (95% CI) |
|------|----------|----------|-----|-------------------|--------------------------|
| var29 | Central | 4 | C | 0.3% (0.1, 0.5) | 27% (22, 32) |
| var27 | Central | 4 | B | 0.8% (0.5, 1.2) | 23% (19, 28) |
| var35 | Central | 4 | C | 3.3% (2.2, 4.7) | 8.0% (5.8, 11) |
| var28 | Central | 4 | C | 2.2% (1.4, 3.1) | 6.9% (5.2, 9.1) |
| var32 | Central | 4 | C | 4.7% (2.5, 5.9) | 2.5% (1.5, 3.3) |
| var36 | Central | 4 | C | 1.8% (0.2, 4.2) | 2.5% (1.5, 4.1) |
| var31 | Central | 4 | C | 1.4% (0.9, 2.0) | 2.2% (1.7, 2.9) |
| var34 | Central | 6 | C | 1.3% (0.1, 3.4) | 2.1% (1.4, 3.6) |
| var13 | Subtelomeric | 4 | B | 1.9% (1.5, 2.4) | 2.1% (1.6, 2.8) |
| var30 | Central | 4 | C | 1.9% (1.3, 2.7) | 1.9% (1.4, 2.5) |
| var10 | Subtelomeric | 5 | B | 3.4% (1.1, 5.7) | 1.7% (0.94, 2.7) |
| var17 | Central | 6 | B | 3.0% (0.8, 5.6) | 1.7% (0.90, 2.8) |
| var24 | Central | 6 | B | 1.9% (0.1, 5.2) | 1.6% (0.85, 3.1) |
| var7 | Subtelomeric | 7 | B | 2.2% (0.3, 5.1) | 1.5% (0.83, 2.7) |
| var19 | Central | 4 | B | 1.3% (0.1, 3.2) | 1.4% (0.93, 2.5) |
| var26 | Central | 4 | B | 1.5% (0.2, 3.4) | 1.4% (0.90, 2.3) |
| var25 | Central | 6 | B | 2.7% (0.4, 5.6) | 1.2% (0.60, 2.1) |
| var1csa | Subtelomeric | 8 | A | 5.5% (4.4, 6.0) | 1.2% (0.89, 1.5) |
| var33 | Central | 4 | C | 2.7% (0.5, 5.8) | 1.1% (0.57, 1.9) |
| var16 | Subtelomeric | 4 | B | 2.8% (1.0, 5.3) | 0.91% (0.52, 1.5) |
| var4 | Subtelomeric | 8 | A | 4.3% (2.1, 5.9) | 0.86% (0.52, 1.3) |
| var14 | Subtelomeric | 4 | B | 4.0% (2.2, 5.9) | 0.73% (0.45, 1.1) |
| var22 | Central | 7 | B | 1.9% (0.2, 5.1) | 0.72% (0.42, 1.3) |
| var21 | Central | 7 | B | 4.7% (2.7, 5.9) | 0.70% (0.46, 0.94) |
| var8 | Subtelomeric | 7 | B | 3.7% (1.8, 5.8) | 0.66% (0.39, 1.0) |
| var11 | Subtelomeric | 5 | B | 1.5% (0.9, 2.2) | 0.60% (0.45, 0.80) |
| var18 | Subtelomeric | 4 | B | 5.2% (3.8, 6.0) | 0.60% (0.42, 0.79) |
| var5 | Subtelomeric | 6 | A | 3.5% (2.7, 4.3) | 0.53% (0.39, 0.69) |
| var39p† | Subtelomeric | 2 | B | 4.7% (2.7, 6.0) | 0.48% (0.30, 0.66) |
| var50Ψ‡ | Central | 6 | C | 4.4% (2.0, 5.9) | 0.46% (0.26, 0.67) |
| var23 | Central | 6 | B | 0.6% (0.0, 2.0) | 0.38% (0.26, 0.55) |
| var9 | Subtelomeric | 6 | B | 2.8% (0.7, 5.3) | 0.36% (0.20, 0.58) |
| var2csa | Subtelomeric | 6 | E | 4.7% (2.3, 5.9) | 0.28% (0.16, 0.40) |
| var2 | Subtelomeric | 6 | A | 1.4% (0.2, 4.2) | 0.27% (0.17, 0.50) |
| var12 | Subtelomeric | 4 | B | 3.8% (1.5, 5.8) | 0.26% (0.15, 0.40) |
| var20 | Subtelomeric | 4 | B | 4.1% (1.8, 5.9) | 0.25% (0.14, 0.37) |
| var1 | Subtelomeric | 7 | A | 5.5% (4.2, 6.0) | 0.024% (0.018, 0.031) |
| var6 | Central | 8 | A | 2.9% (0.4, 5.9) | 0.018% (0.0086, 0.032) |

*Number of encoded DBL (Duffy binding-like) and CIDR (Cys rich inter-domain region).
†Partial gene.
‡Psuedogene.

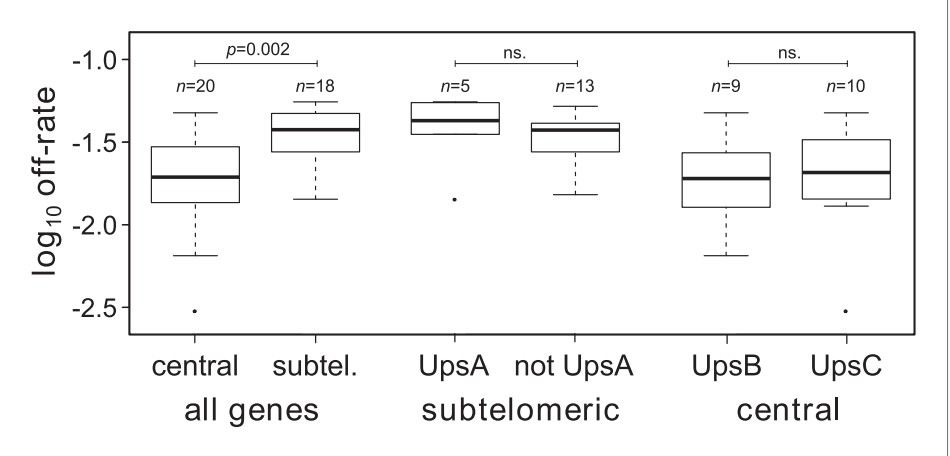

**Figure 3**. Off-rate estimates show strong dependency on chromosomal location. The estimated rates at which active var genes become silenced are significantly lower for centrally located var genes than for those in subtelomeric locations. There was no significant effect of upstream promoter type when differentially testing off-rates of genes in subtelomeric location (UpsA vs non-UpsA) or in central location (UpsB vs UpsC).

confirmed that activation biases of centrally located *var* genes are significantly higher than those of subtelomeric genes (**Figure 4**, Welch's $t$ = 2.8, p=0.009; accounting for uncertainty p=0.011). In fact, of the highest four activation biases, three belonged to neighbouring genes on chromosome 4: *var29*, *var27* and *var28*. Importantly, no significant and independent effect of promoter type was found, which again points towards a strong influence of chromosomal location on *var* gene activation.

Additionally, we considered whether gene architecture had an effect on the observed activation biases, as recent research suggests that gene length, in terms of the number of binding domains, is another important *var* gene characteristic besides chromosomal location and promoter type (**Buckee and Recker, 2012**). Grouping genes depending on whether they encoded only four domains (short

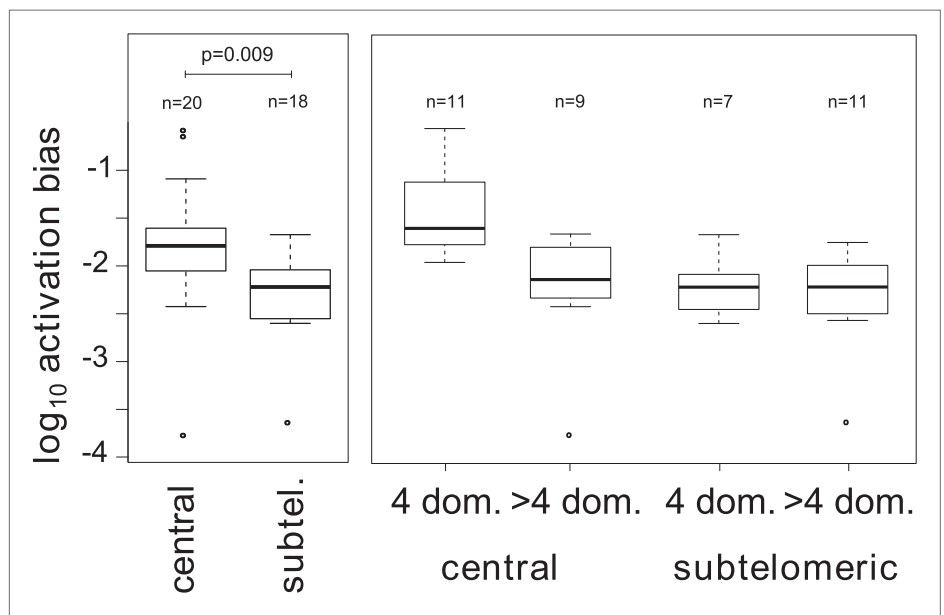

**Figure 4**. Associations between activation biases and genetic attributes. The mean activation bias is higher for centrally located *var* genes. Within the set of centrally located genes, those encoding only four binding domains have higher activation biases than longer genes. Together, short, central genes have activation biases ≈ 10 times higher than the rest of the repertoire.

genes) or more than four domains (long genes) showed that activation biases differed significantly depending on gene length (*Figure 4*, $F = 9.7$, p=0.004) as well as chromosomal location ($F = 8.1$, p=0.007) and the interaction of these two factors ($F = 5.0$, p=0.032). As such, central *var* genes encoding only four binding domains (also known as Type 1 *var* genes [*Gardner et al., 2002*]) had on average a 10 times higher activation bias than the rest of the repertoire. Results were similar when we accounted for uncertainty in our estimates (p=0.005, p=0.0096, p=0.037, respectively).

### *Var* gene switching is influenced by the deactivating gene

We next tested explicitly whether the currently active gene has an influence on the direction of switching, that is if in general $\beta_{ij} \neq \beta_{kj}$. We used an optimisation algorithm ('Materials and methods') to fit two alternative models to the data for the 16 most highly transcribed *var* genes: (i) a general model including off-rates and individual switch biases (240 independent parameters), and (ii) a simpler model, in which all sets of switch biases were required to be identical, that is where $\beta_{ij} = \beta_j$ for all $i$ and $j$ (31 independent parameters). A likelihood ratio test indicated that the model with individual switch biases was significantly more likely than the second model ($F = 1.84$, p<0.0001), confirming that antigenic switching in *P. falciparum* is not only controlled by an activation hierarchy but also influenced by the currently active gene. We again tested for associations with chromosomal location and promoter type ('Materials and methods'), which indicated that switch biases were higher between *var* genes with matching chromosomal locations than between differently located genes (*Figure 5*; Welch's $t = 2.2$, p=0.029), whereas no significant association was found between switch biases and matching Ups types or gene lengths. This result needs to be confirmed with larger data sets, however, as statistical significance was lost when we accounted for uncertainty in our parameter estimates.

### *Var* gene sequences suggest a role of activation bias in antigenic evolution

*Var* gene sequences can be described using a set of 628 conserved homology blocks, with an average length of 19 amino acids (*Rask et al., 2010*). Using homology blocks found only within the DBL1 and CIDR1 domains, which are present in all HB3 *var* genes except *var1csa*, *var2csaA*, *var2csaB* and the partial gene *var39*, we constructed a *var* gene homology network, in which edges between genes are defined and weighted by the number of shared homology blocks (*Figure 6—figure supplement 1*). We then calculated the centrality of each gene within this network. This is a distance measure between all pairs of nodes (i.e., genes) and can therefore be used as a proxy for the relatedness of a gene to the rest of the repertoire. The inclusion of network centrality as an explanatory variable in the previous ANOVA model confirmed that the degree by which *var* genes share sequence blocks is positively correlated with activation bias, independently of chromosomal location and gene length (*Figure 6A*; $F = 8.8$, p=0.006 for the model comparison; accounting for uncertainty p=0.006).

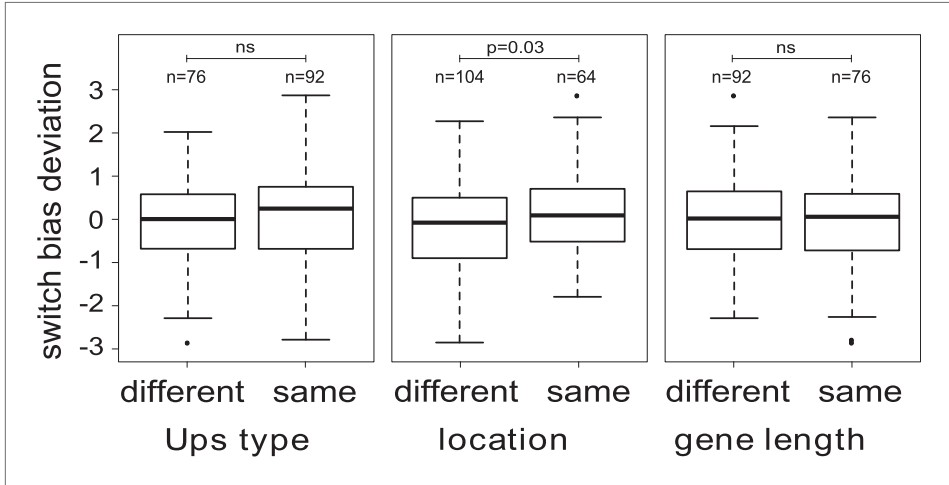

**Figure 5**. Variation in switch biases (deviation from the mean). Antigenic switching between *var* genes is more frequent between genes with matching chromosomal locations, for example from central to central or from subtelomeric to subtelomeric. No significant associations between switch bias and matching Ups type or gene length are found.

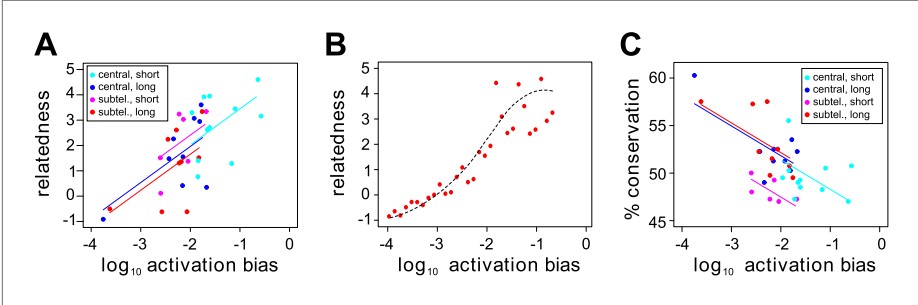

**Figure 6**. Effect of *var* gene activation on sequence evolution. (**A**) *Var* gene activation biases are significantly correlated with each gene's relatedness to the rest of the HB3 repertoire, as measured by the gene's centrality within a shared homology block network, independently of chromosomal location or gene length. Trend lines show the fit of a linear regression model. (**B**) Simulation of *var* gene evolution by gene conversion, whereby homology blocks are swapped among pairs of genes chosen at random according to their activation biases, shows a similar correlation between the genes' activation biases and their relatedness to the rest of the repertoire. The red points show the outcome of one simulation, and the dashed line is the smoothed average of 50 runs. (**C**) Activation biases are negatively correlated with the gene's average domain sequence conservation, as a measure of population-level diversity, independently of chromosomal location and gene length. Trend lines show the fit of a linear regression model.

The following figure supplements are available for figure 6:

**Figure supplement 1**. Relatedness network of the HB3 repertoire.

We hypothesised that this could indicate a possible role of activation bias, or gene transcription, in *var* gene sequence evolution. Under the assumption that gene transcription facilitates recombination we would expect more frequently activated genes to share more sequence blocks with other genes, simply because they more often act as recombination sites. To test this we explicitly simulated gene conversion, in which pairs of genes were chosen at random, weighted by their activation bias and homology blocks copied between them ('Materials and methods). As expected, we found a strong positive correlation between activation bias and our measure of relatedness in terms of network centrality (*Figure 6B*).

Under the same assumption we would also expect genes with higher activation biases to be overall more diverse. Indeed, using the average sequence conservation of all domains as a measure of population level diversity (*Buckee and Recker, 2012*), we again found this to be significantly and positively correlated, independently of chromosomal location and gene length (*Figure 6C*; $F$ = 13.6, p=0.0009 for the model comparison; accounting for uncertainty p=0.0006), suggesting a role of active gene transcription in generating antigenic diversity in *P. falciparum* malaria.

## Discussion

Several studies have noted that *var* genes located near the centres of chromosomes tend to be more highly transcribed in vitro than those in subtelomeric location (*Frank et al., 2007*; *Peters et al., 2007*; *Enderes et al., 2011*; *Zhang et al., 2011*; *Fastman et al., 2012*), and it has been suggested that this might be due to low deactivation rates (*Frank et al., 2007*). In this study, we found that high transcription levels are confined to a subset of central genes that encode only four PfEMP1 binding domains (termed Type 1 *var* genes [*Gardner et al., 2002*]) and, importantly, are the result of high activation biases rather than low off-rates. That is, even though some of the most highly transcribed genes had very low off-rates, the total sum of other genes switching off with high bias towards these genes significantly outweighs the effect of off-rates. This is consistent with our previous study using artificial transcription data, which also showed that off-rates, at least within a biologically plausible range, can have only a modest effect on the total transcript levels within parasite populations (*Noble and Recker, 2012*).

Reanalysis of data from a recent study on the NF54 parasite strain by *Fastman et al. (2012)* supports these findings. Consistent with our results for HB3, centrally located *var* genes encoding only four binding domains were more highly transcribed than the rest of the repertoire (Welch's $t$ = 2.3, p=0.02), with PF3D7_0809100, PF3D7_0421300 and PF3D7_0800100 being the most highly transcribed genes. According to our model, these are predicted to have high activation bias, and indeed, PF3D7_0809100 has previously been reported as the most transcribed *var* gene in NF54 bulk culture, whereas PF3D7_0421300

and PF3D7_0800100 were found to be the two genes most commonly activated by *var* gene switching in previous studies (*Dzikowski et al., 2006*; *Frank et al., 2007*; *Enderes et al., 2011*). Also in agreement with our results, NF54 gene transcription levels were positively correlated with relatedness to the rest of the repertoire in terms of shared DBL1 and CIDR1 homology blocks (Spearman's $\rho$ = 0.3, p=0.044).

Our results further indicate that the direction of transcriptional switching between *var* genes depends not only on intrinsic activation biases but also on the particular preferences of each deactivating gene. Even though these variations were found to have only a modest effect on in vitro switching, they are expected to play a much greater role during infection, where factors such as immune responses and binding affinities can significantly alter the *var* gene expression profiles. Furthermore, the combination of an activation hierarchy and these specific variations could explain our previously proposed source-sink model underlying antigenic switching in *P. falciparum* (*Recker et al., 2011*), in which genes with high in-degrees, i.e., those most commonly switched to, correspond to the genes here identified as having high activation biases. Intriguingly, also, our results suggest that switch biases may be stronger between genes in similar chromosomal locations: central genes prefer to switch to other central genes, and subtelomeric genes prefer to switch to other subtelomeric genes. More data are needed to fully establish the factors influencing variation in switch biases, however, especially for less activated genes, where parameter estimations are more prone to uncertainties.

Despite recent advances in elucidating the genetics and epigenetics of *var* gene activation and silencing, not much is known about how mutually exclusive switching between the 60 *var* genes is controlled at the molecular level. The discovery of intrinsic *var* gene activation biases and their associations with chromosomal locations and other genetic features could therefore have important implications for future studies. These factors are clearly correlated, however, and it is uncertain how they are linked by cause and effect. For example, in a recent study it was found that gene length is positively correlated with sequence conservation (*Buckee and Recker, 2012*), and long *var* genes are more often positioned in subtelomeric regions. Nevertheless, the correlation between activation bias and relatedness, which was found to be independent of chromosomal location, gene length and upstream promoter, suggests a role of gene transcription in facilitating recombination and therefore in generating and maintaining antigenic diversity in *P. falciparum*. Transcription-associated recombination has been observed in other eukaryotes and may result from increased accessibility to recombination proteins or from stalled replication forks (*Prado and Aguilera, 2005*; *Gottipati and Helleday, 2009*). Importantly, for *P. falciparum* this phenomenon could help to explain why different groups of *var* genes within the repertoire display very different levels of diversity within the population (*Buckee et al., 2009*; *Buckee and Recker, 2012*).

It is further possible that some *var* genes are under balancing selection, whereby the relative sequence conservation of longer genes could be due to optimised functionality, for example in terms of binding avidity and/or affinity. This assumption would also help to reconcile apparent contradictions between our observations and *var* gene transcription in vivo. That is, various in vivo studies have reported UpsA *var* genes to be over-expressed during infections in individuals with little previous exposure (*Jensen et al., 2004*; *Bull et al., 2005*; *Lavstsen et al., 2005*; *Kyriacou et al., 2006*; *Warimwe et al., 2009*), despite their low activation rates in vitro. Each *var* repertoire consists of a set of relatively long and conserved genes, such as those predominantly found with in the UpsA group, and another, short and diverse set of genes, such as those predominantly found within the UpsB and UpsC groups (*Buckee and Recker, 2012*). Under the assumption that more conserved genes have an in vivo survival advantage, possibly due to higher binding affinities (*Avril et al., 2012*; *Claessens et al., 2012*; *Lavstsen et al., 2012*), these variants will dominate early infections in naive hosts, regardless of activation rates. Their lower sequence diversity, however, will then lead to a more rapid acquisition of protective immunity, causing a transition towards the expression of more diverse variants, facilitated by their inherent activation biases. In light of this we would also speculate that the proposed order in which individuals acquire immunity (anti-UpsA followed by anti-UpsB/C), as well as the proposed associations between the expression of UpsA *var* genes and severe disease outcome, especially in young children, is less determined by the Ups group itself and is more a consequence of within-host selection of long, conserved genes, which are often but not exclusively found within the UpsA group.

Together, these results demonstrate how selection pressures operating on multiple ecological scales have shaped the phenotypic plasticity embodied within the antigenic repertoire of *P. falciparum*. Within this setup, *var* genes and the molecular mechanisms that underlie their sequential activation and silencing have co-evolved to allow the parasite to express the most advantageous phenotype in response to its current environment.

# Materials and methods

## Parasite culture and cDNA preparation

Parasites were cultured and sorbitol-synchronised using standard techniques (*Trager and Jensen, 1976*; *Lambros and Vanderberg, 1979*), and RNA was extracted from saponin lysed, mid-to-late ring stage parasites with Trizol as previously described (*Kyes et al., 2000*). For cDNA preparation, 5 µg RNA was treated in 20 µl reactions with TurboFree DNase (Ambion®; following manufacturer's recommendations except incubation was at 16°C for 40 min). All DNase treated RNA samples were tested by PCR with primers to fructose-bisphosphate aldolase (PF14_0425) (*Salanti et al., 2003*) to ensure complete removal of gDNA. Duplicate 20 µl reactions consisting of 6 µl RNA were reversed transcribed with 100 ng random hexamers (Invitrogen) and Bioscript reverse transcriptase (Bioline; following manufacturer's instructions) at 40°C for 40 min. The resulting cDNA was stored at −20°C in single-use aliquots and diluted 1:6 for real time PCR analysis. cDNA was synthesised in triplicate (84% of all data points) or duplicate, and real time PCR was done in duplicate for each primer pair on each cDNA.

## Primer design and initial testing

Primers specific for the HB3 *var* exon 1 repertoire were optimised to meet several basic criteria: that they distinguished between the known *var* gene sequences within the HB3 genome; that they were specific for HB3 *var* genes with no cross-reactivity to 3D7 *var* genes (for future application to studies on 3D7 × HB3 cross progeny); and that they reflected measurement of full-length *var* transcript. Primer pairs were designed using Primer3plus (*Kraemer et al., 2007*; *Untergasser et al., 2007*), with parameters manually set for product size range (100–150 nt), primer length (20 nt), Tm (60°C), GC clamp (1 nt), salt correction formula and thermodynamic parameters (SantaLucia 1998); all other parameters were default settings (see *Table 3* for primer sequences). To avoid general cross-detection of all *var* types, the first kilobase of exon 1 containing the conserved DBLα domain was avoided. The final kilobase was also excluded to avoid sterile transcripts (*Su et al., 1995*; *Kyes et al., 2007*). Each primer pair was specific for a single gene except, unavoidably, the *var3* primer pair and the two pairs designed to the highly similar copies of *var2csa*. Each HB3 primer pair was tested five times on gDNA; pairs that varied more than ±1.5 threshold cycles (Ct) values from the median or had efficiency value (Amplification) less than 1.85 were redesigned. Primer pairs that amplified 3D7 genomic DNA were discarded and redesigned so that they were specific for HB3 *var* genes only.

## qRT-PCR

Quantitative real time PCR (qRT-PCR) reactions were prepared with a Corbett CAS1200 liquid handling system and run on a Corbett Rotor-Gene 6000. Each 10 µl reaction contained 1 µl template, 5 µl SensiMix SYBR (Bioline), 3 µl water and 1 µl primer mix (1.6 µM of each primer). The cycling conditions were as follows: hold, 95°C/10 min; 45 cycles (95°C/25 s, 58°C/25 s, 68°C/30 s); melt, 55°C−99°C. Amplicons were checked for a single product of the expected size by agarose gel electrophoresis. Melt analysis confirmed single products.

## Data collection, processing and verification

Transcription levels for each replicate were calculated by the comparative quantification method of *Pfaffl (2001)*. Comparative quantification results are given as expression levels relative to a control (housekeeping) gene, seryl-tRNA synthetase (Seryl-tRNAF agctacctcagaacaaccattatgtgc, Seryl-tRNAR atcctttccatgtgcccctgc), which is arbitrarily set at 1.0. Minor differences in Take Off values between primer pairs were accounted for using a correction factor derived from five replicates on gDNA. Expression was determined using $A^{T.O.-CF}$, where $A$ is the average amplification, $T.O.$ is the average Ct and $CF$ is the correction factor. To express results relative to the control gene, the following formula was used: $A_{ref}^{TO-CF}/A_x^{TO-CF}$. Expression levels were converted to proportions before calculating geometric means of the replicates. Replicates were excluded as outliers (1.1%) if they deviated from the mean by a factor of more than $10^{1.5}$ or had an amplification efficiency below 1.5 (most excluded measurements met both criteria).

Dominant *var* gene expression detected by qRT-PCR was verified as both full-length and dominant by northern blot analysis, comparing specific probes for each dominant gene vs a generic exon 2 probe (data not shown). The primer pairs for *var* genes 5, 11, 13, 27, 28, 29, 31 and 35 were thus verified as specific for detecting full-length transcripts. The *var3* primer pair, which was known to also detect *var5* and pseudogene *var46*, indicated dominant *var3* alongside *var5* in culture #4. However, we were unable to

**Table 3.** HB3 qPCR primers and cross-referenced *var* gene identifiers

| Gene name | Contig name | Broad locus name | F oligonucleotide | R oligonucleotide |
|---|---|---|---|---|
| *HB3 var16* | HB3-1000-1 | PFHG_03232.1 | ccctgtccacaaccatcagc | cgtcgtcgtcatcagtgtcc |
| *HB3 var1* | HB3-1000-2 | PFHG_03234.1 | ccaaaggagaaggcaccacc | acctatggcacccctctcac |
| *HB3 var12* | HB3-1040 | PFHG_03416.1 | gatgctacaaccaccccacc | gtgttaccactcgcccactc |
| *HB3 var27* | HB3-1704_1 | PFHG_03476.1 | gctcccaaccaccacgttcc | gcttcctgctggtggctgtc |
| *HB3 var28* | HB3-1074-2 | PFHG_03478.1 | gatggcacaaaagttggcgg | tgttctgggtcgacctcctc |
| *HB3 var29* | HB3-1074-3 | PFHG_03480.1 | aagaagatggcgacgaaggc | tccggtgatccctcttctgg |
| *HB3 var13* | HB3-1107 | PFHG_03516.1 | tggtaaatgcaagggtgatacagg | tgcatcgttatcactcaccagc |
| *HB3 var1csa* | HB3-1108 | PFHG_03521.1 | cgcaatatgcaactaatgac | acttggcaatattctgaacg |
| *HB3 var2* | HB3-1210 | PFHG_03840.1 | cgaggacaccacggaggagg | ttggtgctgctggttgtggc |
| *HB3 var5* | HB3-1235 | PFHG_03671.1 | aggtctgctccttcagatgcgtg | tgttttccctaccatgacaaggatgcc |
| *HB3 var36* | HB3-1296 | PFHG_04012.1 | atggacaaatgatggtaagg | taggagtaggtgttgcgttc |
| *HB3 var17* | HB3-1296-2 | PFHG_04014.1 | agatggcgacaaaggccaag | ttgggtttggcaccactagc |
| *HB3 var35* | HB3-1296-3 | PFHG_04015.1 | aaacggaaaacctggcctcc | tcgtcttggcctttggcttc |
| *HB3 var14* | HB3-1308 | PFHG_04035.1 | ggtggtggtgccgatcccgcc | tgtgacgcctccgtcttagtggccc |
| *HB3 var8* | HB3-1334 | PFHG_04081.1 | ggcggtgtctgtattccacc | gctgcctcaccacctgttag |
| *HB3 var9* | HB3-1408 | PFHG_04057.1 | tgctatgacgtgtaatgcccc | acttacatgagtccatctggtg |
| *HB3 var32* | HB3-1459 | Fbadf | ggaaaccgcggtggactcac | acttgtgggtgctttggggc |
| *HB3 var11* | HB3-1499 | PFHG_04491.1 | attggatgatgcctgtcgcc | ggcacctggtttagtggtgg |
| *HB3 var19* | HB3-1514 | PFHG_04620.1 | aaactgacaatggcccccgac | gttgttgagggggtcttcgg |
| *HB3 var18* | HB3-1523 | PFHG_04593.1 | gcggctcacccgacatcttc | gccgcctcgtcttcttcgtc |
| *HB3 var10* | HB3-1587 | PFHG_04749.1 | accactcgtgccaccacctc | gagtttgtacctggcaccccc |
| *HB3 var7* | HB3-1604-1 | PFHG_04769.1 | agcgagtggtactcaggagg | gatggaccacgagatgtgcc |
| *HB3 var20* | HB3-1604-2 | PFHG_04770.1 | acgaagaagacgatgccacc | gaagtcttcggagcgaccac |
| *HB3 var4* | HB3-1703 | PFHG_04861.1 | tggtgccaaagacccctccc | ggccactcgctgtgtctgtg |
| *HB3 var2csaA* | HB3-1727 | PFHG_05046.1 | gggggaaatgtggggtgccg | gggggatacccacactcattaccag |
| *HB3 var3* | HB3-1737 | PFHG_05052.1 | aaagtgcgaagcacctcccc | cgccactgcagggattagctg |
| *HB3 var2csaB* | HB3-1817 | PFHG_05155.1 | tggtacagctgatggtggtacttccg | tgtgcccgctttacggtttcg |
| *HB3 var39p\** | HB3-2007 | efbe | agccattacgtgcgaagctgg | agcggcacatcggcatttttg |
| *HB3 var34* | HB3-209 | PFHG_00592.1 | agtggtgctgtagagccaaaagac | cctgcggcggtgctgtaagg |
| *HB3 var23* | HB3-699-1 | PFHG_02272.1 | gaacccttgacgacgacac | ctcaacacacgtcaaaggcg |
| *HB3 var30* | HB3-699-2 | PFHG_02273.1 | aagacgacaaacctggcacc | gtcgttgcttttggcttcgg |
| *HB3 var6* | HB3-699-3 | PFHG_02274.1 | attcacagcactgaaagtcc | tcacaatcattaaaagcatcc |
| *HB3 var26* | HB3-699-4 | PFHG_02276.1 | aagcagctgatggaacggac | tggttgttgtgggtcttggc |
| *HB3 var31* | HB3-699-5 | PFHG_02277.1 | cgcgaagacgaaaacgtcac | gtttcatccggaccgtcctc |
| *HB3 var50Ψ†* | HB3-752-1 | PFHG_02419.1 | tggtaatgatgaagatgacg | gaattggcttcactttgttc |
| *HB3 var24* | HB3-752-2 | PFHG_02421.1 | gctcgctctttaccacccgc | ttccgtctcctccttcgccg |
| *HB3 var25* | HB3-752-3 | PFHG_02423.1 | agtggtgccaaaactgtcgg | accacaaaagtcgcttcccc |
| *HB3 var22* | HB3-752-4 | PFHG_02425.1 | ccaccacaaaaccccctccag | tccgcttgtggttcgtcttc |
| *HB3 var33* | HB3-752-5 | PFHG_02429.1 | acagaaagttggacaggatg | atggttgttttgagaattgc |

\*Partial gene.
†Pseudogene.

detect *var3* expression by northern blot. As this sample was positive by northern blot probed with *var5*, and it was the only one with 'dominant' *var3* expression by qRT-PCR, we discarded all *var3* qRT-PCR data as ambiguous. We also discarded all the data from one of the two *var2csaA/var2csaB* primer pairs.

## Parameter estimation

Following *Recker et al. (2011)*, we assumed that *var* transcription profiles can be described by a time-discrete model:

$$v_{i,c,t+1} = (1 - \omega_i) v_{i,c,t} + \sum_{j \neq i} \omega_j \beta_{ji} v_{j,c,t},$$

where $v_{i,c,t}$ is the relative transcript level of variant $i$ in culture $c$ at time $t$, $\omega_i$ is the variant specific off-rate and $\beta_{ji}$ is the switch bias (probability of a switch) from variant $j$ to variant $i$. We assumed a maximum off-rate of 6% per generation, which is consistent with previous studies and theory (*Roberts et al., 1992*; *Horrocks et al., 2004*; *Frank et al., 2007*; *Recker et al., 2011*).

We used a Markov chain Monte Carlo (MCMC) method to obtain posterior distributions of likely values for the switch biases and off-rates, using the pooled data from all observed cultures as input. For model comparisons we used a simulated annealing algorithm to find maximum likelihood parameter sets. These methods have previously been described in detail and tested using diverse artificial data sets (*Noble and Recker, 2012*). For some of our analyses it was desirable to use a model of reduced dimension. Previously we have shown that reliable parameter estimates can be obtained from data for the 16 most highly transcribed *var* genes (*Noble and Recker, 2012*), which in this case comprised the 9 starter genes and the 7 others with highest average transcript levels. These 16 genes together accounted, on average, for more than 92% of the sum of transcript levels in each culture (excluding the culture's starter gene), whereas none of the excluded genes accounted for more than 1%. To estimate activation biases, we modified the model so that the direction of switching was independent of the previously active gene (i.e., $\beta_{ji} = \beta_{ki}$ for all $i$, $j$ and $k$).

Before conducting statistical tests, we transformed switch biases using the logit function $\text{logit}(x) = \log(x/(1-x))$ to map from the range $(0,1)$ to the range $(-\infty, \infty)$. Additionally, to account for uncertainty in our parameter estimates, we performed tests on the collected output of our MCMC method. This method samples from the posterior distribution, which means it selects numerous parameter sets such that the probability of each being chosen depends on how likely it is to explain the data. We performed our statistical tests on all the selected parameter sets and calculated mean p-values. We used the same parameter sets to obtain credible intervals for our estimates, as shown in *Table 2*.

## Variation in switch biases

We found that all *var* genes had similar sets of switch biases, such that $\beta_{ij} \approx \beta_{kj}$ for each $i$, $j$ and $k$. A model comparison indicated that these sets were not identical but varied depending on the deactivating gene. To estimate these variations we took a subset of the MCMC switch bias estimates and standardised each set $\{\beta_{1j}, \beta_{2j}, \ldots, \beta_{nj}\}$ by subtracting the mean and dividing by the standard deviation (after rescaling each set to compensate for $\beta_{jj} = 0$). Negative values then corresponded to biases that were smaller than average and positive values to those that were larger. We used the subset of switch biases from the 9 starter genes to the 25 most transcribed genes, omitting the diagonal terms.

## Relatedness network

We constructed a network in which each node represented a gene and each edge was weighted by the number of shared variable DBL1 and CIDR1 homology blocks. Each gene's centrality within the network was calculated using a weighted closeness measure (*Newman, 2001*) implemented by the tnet package for R (*Opsahl, 2009*). We applied variance stabilising transformations to the data prior to analysis. Specifically, the centrality measure was rescaled and squared, and to each switch bias we applied the logit transformation, added five and then square-rooted the magnitude. These transformations made the data conform better to the assumptions of our statistical model; test results would have been more significant had we instead used the untransformed data.

## Gene conversion model

We simulated gene conversion in a repertoire of 30 genes, each comprising 40 homology blocks, with 30 possible variants for each homology block. Genes were initially constructed of randomly chosen homology blocks and were assigned activation biases according to a geometric sequence. For 30,000

iterations, pairs of genes were selected at random, with the probability of selection being proportional to activation bias. A randomly chosen homology block was copied from the first of the chosen genes to replace the corresponding block in the second gene. Homology blocks in all genes also underwent random mutation at a rate 10 times lower than the gene conversion rate. We constructed a relatedness network and calculated centrality in the same way as for the HB3 data.

## Initial conditions of the model

Our switching model required a set of initial conditions. Assuming that each culture, excluding the parent, began with a single parasite and that populations expanded approximately fivefold per generation, we deduced that the first transcriptional switch most likely occurred, on average, two or three generations after cloning. Therefore we assumed that each of these cultures remained clonal for the first three generations. For the parent culture, we used the first observation as the initial condition. If a switch had occurred in one or more of the initially clonal cultures within the first few generations, when the population was very small, then our assumed initial conditions would be inaccurate. Therefore we also ran our algorithms using the first observations as the initial conditions for all cultures. The parameter estimates were highly consistent with those obtained using clonal initial conditions.

## Cross-validation of parameter estimates

The simplest way to estimate switch biases and off-rates for a single *var* gene is to obtain a clone transcribing the gene, grow it in culture and analyse changes in transcript levels over time. A particular strength of our method was that it also estimated parameters for genes we did not clone. To assess the accuracy of results obtained in this way, we performed cross-validation by sequentially excluding data sets. For example, to derive alternative estimates for *var29* we ran the MCMC algorithm without the data for culture #8 (which initially expressed *var29*), and to derive alternative estimates for *var27* we excluded data for cultures #5 and #6. The results, shown in *Figure 2—figure supplement 1*, confirmed our method's ability to estimate switch biases of non-starter genes with sufficient accuracy to determine the general switching pattern. Off-rate estimates were however not always accurate, especially for genes with very low transcript levels.

We used a similar method to test the accuracy of our method for estimating variations in switch biases. Cultures #2 and #3 initially expressed the same starter gene *var13*, and both cultures #5 and #6 initially expressed *var27*. By excluding the data from culture #2 we derived estimates for the *var13* switch biases mostly dependent on the culture #3 data, and vice versa. Similarly we excluded data from each of cultures #5 and #6 to obtain two sets of estimates for the *var27* switch biases. For each of the two different data sets we calculated the deviations from the mean switch biases and found these to be positively correlated (Pearson's $\rho$ = 0.43, p=0.015), confirming the validity of our method.

## Validation of model assumptions

Our likelihood function, used for parameter estimation and model comparisons, assumed that input data were subject to measurement errors following a log-normal distribution. We have previously shown that our MCMC algorithm reliably determines network structures when the measurement error distribution is log-normal (base 2) with standard deviation $\sigma \leq 1$, and that $\sigma$ can be reliably estimated using our simulated annealing algorithm (*Noble and Recker, 2012*). In the 16-gene model, $\sigma \approx 0.73$, which was well within the required range.

To test whether the measurement errors obeyed a log-normal distribution, we analysed the distribution of the deviations between the data and the simulated annealing output, after applying log transformations. The distribution for the 16-gene model was unimodal and only slightly asymmetrical (skewness 0.4) but was more peaked than a normal distribution (excess kurtosis 2.4), so that it somewhat resembled a Laplace distribution. We tried running the simulated annealing algorithm with an error function that assumed log-Laplace-distributed noise, and this made very little difference to the results. The deviations for the constrained 38-gene model were similarly distributed, excluding a very small minority of outliers.

Unusually large deviations occurred at 13 of the 64 observed time points, 8 of which were between generations 48 and 52 and therefore belonged to the same round of measurements. Transcript levels deviated more for longer genes than for shorter genes, and northern blotting of the samples taken at the affected time points showed unusually low levels of long, for example *var* length, RNA, confirming that the cause was not PCR error but most likely an as yet unidentified artefact of RNA preparation. Excluding the 13 data points affected by this phenomenon made very little difference to our parameter estimates and statistical test results.

Analysis of the qRT-PCR replicates revealed that measurements of lower transcript levels had higher standard errors, but the average standard error increased only threefold for each 1000-fold difference in transcript level.

## Growth rates

It has been suggested that *var* gene transcript levels in culture may be affected by small differences in parasite growth rates (*Horrocks et al., 2004*), although no evidence has been found to support this hypothesis. To test whether such differences might affect our estimates we added to our model an excess growth rate parameter $\phi_i$ for each gene $i$:

$$v_{i,c,t+1} = (1 - \omega_i)(1 + \phi_i)v_{i,c,t} + \sum_{j \neq i} \omega_j \beta_{ji}(1 + \phi_j)v_{j,c,t}.$$

When we allowed growth rates to vary by up to 0.5% per generation (equivalent to approximately 50% over the duration of the experiment), the extended model was not significantly more likely than the model with only switch biases and off-rates. When we increased the upper bound on the excess growth rates to 1% per generation, the extended model became significantly more likely. However, such large differences in growth rates in vitro are biologically implausible and have never been observed. Importantly, the inclusion of growth rate differences had very little effect on our switch bias and off-rate estimates.

## Acknowledgements

We would like to thank Caroline O Buckee for providing estimates of *var* gene domain sequence conservation. The work was funded by the Wellcome Trust (Grant No. 082130/Z/07/Z to CN), the Biotechnology and Biological Sciences Research Council (studentship to RN) and the Royal Society (University Research Fellowship to MR).

## Additional information

### Funding

| Funder | Grant reference number | Author |
| --- | --- | --- |
| Wellcome Trust | 082130/Z/07/Z | Chris I Newbold |
| BBSRC | | Robert Noble |
| Royal Society | | Mario Recker |

The funders had no role in study design, data collection and interpretation, or the decision to submit the work for publication.

### Author contributions

RN, Analysis and interpretation of data, Drafting or revising the article; ZC, Acquisition of data, Analysis and interpretation of data; SK, Acquisition of data, Analysis and interpretation of data, Drafting or revising the article; RP, Acquisition of data, Analysis and interpretation of data; CIN, MR, Conception and design, Analysis and interpretation of data, Drafting or revising the article

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
