## [Decision Letter]

Thank you for sending your work entitled ‘The antigenic switching network of *Plasmodium falciparum* and its implications for the immuno-epidemiology of malaria” for consideration at *eLife*. Your article has been favorably evaluated by a Senior editor, a Reviewing editor, and 3 reviewers.

The Reviewing editor and the reviewers discussed their comments before we reached this decision, and the Reviewing editor has assembled the following comments to help you prepare a revised submission.

This well written manuscript describes a thorough in vitro study of the *var* expression pattern in the *Plasmodium falciparum* HB3 laboratory strain. The authors study 11 clones for 98 to 108 generations of growth and assess transcription in each culture at 4–7 time points. The relative transcriptional signal (as a percentage of the total *var* gene transcriptional signal) was documented for every measurement and the resulting transcriptional profile was analyzed according to chromosomal position, promoter type, gene length, and sequence diversity. The authors find an expression hierarchy favoring centrally located *var* genes, as has previously been found in other parasite isolates. Specifically centromeric type 1 *var* are found to be highly expressed due to high activation rates. Furthermore it is noticed that frequently expressed *var* genes are more genetically diverse, leading the authors to suggest that *var* transcription facilitates generation of antigenic diversity.

The strength of this study lies mainly in the high quality dataset and the previously published modeling methodology applied to it, giving a detailed quantitative characterization of the *var* expression hierarchy.

The referees agree that all results are experimentally and statistically very solid. They raise a few major points that follow:

1) One question that is not directly evaluated is that of low off-rates of UpsC genes. UpsC genes are generally located in central areas of chromosomes. Therefore this promoter type could potentially serve as a marker for central *var* genes in parasite isolates in which chromosomal position of individual *var* genes is not known. It would therefore be of great benefit to the field if the authors could use their data to ask the question if UpsC promoter off-rates are always very low. For example this data point could be added to the first part of Figure 2 (all genes) by indicating the promoter types of the central and subtelomeric *var* genes (central: UpsC, UpsB and Ups A; subtelomeric: Ups B and UpsA). The authors are in a unique position to ask this question and potentially could provide the field with guidance on how to assess central *var* gene transcriptional activity in field isolates.

2) The effect of parameter estimation uncertainty on the findings should be clarified. The Discussion mentions that the local switch bias parameter estimates are uncertain (also shown with fuzziness in Figure 3). It must be possible to calculate the uncertainty of the activation-rate and off-rate parameters in Table 2, and these should be provided.

3) Several of the findings based on the estimated parameters have p-values >=0.01 (e.g., Figures 2, 3 and 4). To determine if these findings are significant despite parameter estimation uncertainty, the effect of such uncertainty on these p-values should be calculated/discussed. Moreover, to give the reader an intuitive idea of the significance of the box plot figures, each box should be accompanied by the number of data points it represents (e.g., in Figure 2: UpsA [N=5]). A numeric matrix with the estimated switch bias parameters and uncertainties (Figure 3) should be made available as supplementary material.

4) The central concept of activation rate should be introduced more clearly. The sentence “This finding led us to consider switch bias as the sum of two factors: a ‘global activation rate’, solely dependent on the gene to be activated, and a ‘local bias’, dependent on both the activated and the deactivated gene”, can be interpreted as: switch bias = activation rate + local bias, where switch bias is the previously defined *β*_*j,i*_, and that this parameter is split in two. It took some time to gather from the remaining text that the activation rate is actually defined as the average switch bias towards a gene.

5) The mentioning of a “global activation hierarchy” is unfortunate (especially in the Abstract) since it is ambiguous and suggests that the findings are valid for parasites worldwide, which is not necessarily the case since this study only concerns the HB3 isolate. The word “global” can be removed without subtracting essential meaning. Similarly, the words “global” and “local” are not necessary when mentioning the parameters “global activation rate” and “local switch bias”.

---

## [Author Response]

We would like to thank the referees and senior editor for their favourable review of our manuscript and their helpful comments and suggestions, which we took into consideration in preparing the revised version. In particular, we have reworded various parts in the Introduction and added many more key references that were missing in the original manuscript. We have also clarified the terminology and made sure we use important terms, such as “activation bias” and “switch bias”, more consistently throughout. Finally, we have conducted additional statistical tests and have added credibility intervals to our parameter estimates. A point-by-point response to specific points is given below.

*1) One question that is not directly evaluated is that of low off-rates of UpsC genes. UpsC genes are generally located in central areas of chromosomes. Therefore this promoter type could potentially serve as a marker for central* var *genes in parasite isolates in which chromosomal position of individual* var *genes is not known. It would therefore be of great benefit to the field if the authors could use their data to ask the question if UpsC promoter off-rates are always very low. For example this data point could be added to the first part of Figure 2 (all genes) by indicating the promoter types of the central and subtelomeric* var *genes (central: UpsC, UpsB and Ups A; subtelomeric: Ups B and UpsA). The authors are in a unique position to ask this question and potentially could provide the field with guidance on how to assess central* var *gene transcriptional activity in field isolates*.

UpsC genes are exclusively positioned in central chromosomal location and we investigated the difference between UpsC genes and centrally located UpsB genes (there is only one central UpsA gene in HB3). The results show that there was no significant difference in the off-rates of UpsC genes compared to UpsB genes (shown in Figure 2).

*2) The effect of parameter estimation uncertainty on the findings should be clarified. The Discussion mentions that the local switch bias parameter estimates are uncertain (also shown with fuzziness in Figure 3). It must be possible to calculate the uncertainty of the activation-rate and off-rate parameters in Table 2, and these should be provided*.

We have added 95% credible intervals to Table 2, calculated from all the parameter sets selected by our MCMC method. As a result of this process, we also refined our point estimates of the switching parameters (as listed in Table 2). We reran all of our statistical tests and redrew our figures in light of these more accurate estimates, though this resulted in only very minor changes.

*3) Several of the findings based on the estimated parameters have p-values >=0.01 (e.g., Figures 2, 3 and 4). To determine if these findings are significant despite parameter estimation uncertainty, the effect of such uncertainty on these p-values should be calculated/discussed. Moreover, to give the reader an intuitive idea of the significance of the box plot figures, each box should be accompanied by the number of data points it represents (e.g., in Figure 2: UpsA [N=5]). A numeric matrix with the estimated switch bias parameters and uncertainties (Figure 3) should be made available as supplementary material*.

We have rerun all our statistical tests using all the parameter estimates selected by our MCMC method. As noted above, we have added credibility intervals to Table 2, and we further provide the number of data points in the box plots.

*4) The central concept of activation rate should be introduced more clearly. The sentence: “This finding led us to consider switch bias as the sum of two factors: a ‘global activation rate’, solely dependent on the gene to be activated, and a ‘local bias’, dependent on both the activated and the deactivated gene” can be interpreted as: switch bias = activation rate + local bias, where switch bias is the previously defined* β_j,i_*, and that this parameter is split in two. It took some time to gather from the remaining text that the activation rate is actually defined as the average switch bias towards a gene*.

We agree that this was a bit too condensed and also not very consistent in its use. In our revised version we provide more detail on how we define switch biases and activation biases and use these more consistently throughout. Specifically, switch bias, *β*_*ij*_, refers to the probability that when gene *i* switches off, gene *j* becomes activated, whereas a gene’s activation bias is the average of all switch biases towards this gene and therefore the per switch activation probability.

*5) The mentioning of a “global activation hierarchy” is unfortunate (especially in the Abstract) since it is ambiguous and suggests that the findings are valid for parasites worldwide, which is not necessarily the case since this study only concerns the HB3 isolate. The word “global” can be removed without subtracting essential meaning. Similarly, the words “global” and “local” are not necessary when mentioning the parameters “global activation rate” and “local switch bias”*.

We agree and have removed all references to global and local biases. Our initial reason for choosing “global activation bias” was to highlight the fact that there appears to be a hierarchy of gene activation that is independent of the currently active gene, i.e. independent of local effects. In this respect global referred to the repertoire as a whole, rather than a universal activation hierarchy, which would apply to all *var* gene repertoires.